# Fifteen-Year Surveillance of LTR Receiving Pre-Emptive Therapy for CMV Infection: Prevention of CMV Disease and Incidence of CLAD

**DOI:** 10.3390/microorganisms10122339

**Published:** 2022-11-25

**Authors:** Davide Piloni, Elisa Gabanti, Monica Morosini, Gabriela Cassinelli, Vanessa Frangipane, Federica Zavaglio, Tiberio Oggionni, Laura Saracino, Sara Lettieri, Eloisa Arbustini, Federica Meloni, Daniele Lilleri

**Affiliations:** 1Pneumology Unit, Internal Medicine and Infectious Diseases Department, Fondazione IRCCS Policlinico San Matteo, 27100 Pavia, Italy; 2Molecular Virology Unit, Microbiology and Virology Department, Fondazione IRCCS Policlinico San Matteo, 27100 Pavia, Italy; 3Pneumology Unit, Fondazione IRCCS Policlinico San Matteo, University of Pavia, 27100 Pavia, Italy; 4Laboratories of Genetics, Transplantology and Cardiovascular Diseases, Fondazione IRCCS Policlinico San Matteo, 27100 Pavia, Italy

**Keywords:** acute rejection, bronchoalveolar lavage, cytomegalovirus, ganciclovir, valganciclovir, lung transplant recipient, chronic lung allograft dysfunction, pre-emptive therapy

## Abstract

The efficacy of pre-emptive therapy in the prevention of cytomegalovirus (CMV) disease and the potential association of CMV infection with the occurrence of chronic lung allograft dysfunction (CLAD) was evaluated in 129 lung transplant recipients receiving pre-emptive therapy based on pp65-antigenemia or CMV-DNA in the blood and in the bronchoalveolar lavage. Seventy-one (55%) patients received pre-emptive ganciclovir/valganciclovir (GCV/VGCV) for CMV infection for a median of 28 (9–191) days. Possible CMV disease occurred in six (5%) patients and was healed after the GCV/VGCV therapy. The cumulative incidence of CLAD was 38% and 54% at 5 and 10 years. Acute rejection and CMV load in the blood (but not in the lung) were independent predictors of the occurrence of CLAD. Pre-emptive therapy is highly effective in preventing CMV disease in lung recipients and does not induce a superior incidence of CLAD compared to what reported for other cohorts of patients who received an extended antiviral prophylaxis.

## 1. Introduction

Lung transplantation (LT) is the best therapeutic option for end-stage lung disease. Despite major improvements in the clinical management of these patients, survival after lung transplantation remains lower than for other solid organ transplant (SOT) recipients. The estimated survival percentage decreases from approximately 80% in the first year to about 50% at 5 years [1]. The major factor from 3 to 5 years after transplant is represented by the development of chronic lung allograft dysfunction (CLAD); on the contrary, during the first year LT failure is mainly due to infections [2], among which cytomegalovirus (CMV) infection is the most important viral complication in SOT recipients [3]. 

CLAD is a heterogeneous condition and represents the main cause of death in LTR. According to a recent classification, there are two major clinical phenotypes of CLAD: the so-called obstructive form, i.e., bronchiolitis obliterans syndrome (BOS) [4], (nearly 70% of cases), and the restrictive allograft syndrome (RAS) [5,6]. BOS affects almost 50% of lung transplant recipients within 5 years after transplant, causing significant morbidity and mortality [4]. In addition, BOS that manifests early after transplantation shows a poorer prognosis than late-onset BOS [4]. RAS is the pathophysiologic presentation of as many as 30% of CLAD cases [4,5]. Moreover, the prognosis of RAS is much poorer than that of BOS [4,5]. The most important risk factors that have been associated with the development of CLAD are recurrent episodes of acute rejection [7] and infections [8,9].

Cytomegalovirus (CMV) infection is the most frequent viral complication in LT recipients. After primary infection, CMV establishes life-long latency, with sporadic subclinical reactivations in healthy immunocompetent subjects. In transplant recipients undergoing immunosuppressive treatment, both primary and reactivated CMV infection can lead to severe diseases. Without some form of prevention, CMV infection primarily occurs in the first 3 months following transplant [10]. The typical clinical manifestations range from persistent fever and mononucleosis-like syndrome to more severe invasive organ disease, such as CMV hepatitis, gastrointestinal disease, and pneumonitis [11]. CMV disease, and in particular CMV pneumonitis, has been associated with loss of graft function and survival in lung transplantation recipients [12]. In the absence of any prophylaxis, the frequency of CMV infection and disease is highly variable among studies, ranging from 38% to 75% [13,14]. Major risk factors for CMV infection/disease are donor/recipient mismatch (D+/R−), [10,15], high-dose corticosteroids [11,16], and induction treatment (mainly anti-lymphocyte agents) [12] or anti-rejection therapy [10,17,18].

Most frequently, anti-CMV therapies rely on intravenous ganciclovir or oral valganciclovir administration. The latter drug is usually chosen as first-line therapy due to its excellent oral bioavailability [14,19]. Two preventive strategies are currently adopted: antiviral prophylaxis and pre-emptive therapy. The antiviral prophylaxis consists in the administration of antiviral drugs to all transplanted patients for 6–12 months. In fact, valganciclovir has proven to be far more effective in preventing CMV infection/disease when administered on a long-term basis (i.e., 6–12 rather than <3 months) [14,20]. A previous survey has revealed that the majority of LT centres administer CMV-specific hyperimmune globulin in combination with antiviral therapy solely in D+/R− patients instead of using it as a part of a universal prophylaxis [14,21,22]. The pre-emptive therapy consists in the administration of antiviral agents after reaching a predetermined cutoff [22,23,24,25,26] of CMV load in the blood and/or bronchoalveolar lavage (BAL), but before the development of clinical symptoms. While both types of management have advantages and side effects, recent studies have pointed out that both prophylaxis protocols and pre-emptive therapy can attenuate the risk of CMV infections and their complications, such as graft failure [27,28].

The antiviral prophylaxis, on one hand, allows the clinicians to reduce the need for strict virological monitoring, while its disadvantages are mainly related to the side effects of antiviral drugs in the kidney and bone marrow as well as to the economic costs [29,30]. The pre-emptive strategy, on the other hand, has a lower impact in terms of the systemic side effects of antiviral drugs but requires a strict control of the CMV load. The pre-emptive therapy may not prevent the indirect effects of CMV infection, including those on graft and patient survival [23]. The main issue is the lack of a universal agreement about the type of specimen (blood and/or BAL) and related cutoffs [29,30].

More recently, the Lausanne group [31] studied LT patients receiving the antiviral prophylaxis for 3–6 months followed by monitoring of CMV replication during the first-year post-transplantation and pre-emptive therapy. In this cohort, CMV infections did not seem to have an impact on long-term allograft lung function.

The objective of the present study was to analyse the impact of pre-emptive therapy in the prevention of CMV disease and the potential association of CMV infection in the blood and in the lungs with the occurrence of CLAD in a large cohort of LT recipients not receiving anti-CMV prophylaxis.

## 2. Materials and Methods

### 2.1. Study Design and Patient Population

We analysed clinical and virological data from 129 patients who received single or double lung transplantation at Fondazione IRCCS Policlinico San Matteo from February 2004 to February 2018. Clinical follow-up in the post-transplantation period was performed at the Pneumology Unit of Fondazione IRCCS Policlinico San Matteo, and virological follow-up at the microbiology and Virology Unit of the same Institution. The data were analysed as per February 2019.

Among the 38 patients not considered for the analysis, 35 died due to early complications (median time 16 days, range 1–75 days) not related to CMV infection, whereas for the other 3 patients, the clinical and virological data were not recorded.

### 2.2. Monitoring and Diagnosis of CMV Infection or Disease

CMV infection in the blood was monitored by CMV DNA quantification in whole blood (*n* = 101) or pp65-antigenemia (*n* = 28) determination, and in the BAL by CMV DNA. In the 28 patients monitored by pp65-antigenemia, CMV DNA in the blood was determined retrospectively: these patients were previously enrolled in a randomized study comparing the two assays as a guide for pre-emptive therapy [24]. CMV DNA was quantified by real-time PCR using a primer pair selected from the US8 gene (nt 226–290). Antigenemia was determined by counting the number of pp65-positive leukocytes/2 × 10^5^ leukocytes isolated from peripheral blood. During the first 3 months after transplantation, HCMV infection was monitored once a week in the case of undetectable infection and twice a week in the case of CMV infection. Subsequently, in the absence of symptoms, the patients were monitored for CMV once a month until day +180, then at days +270 and +360. However, in the presence of CMV infection or specific symptoms, monitoring for HCMV infection followed the same schedule as during the first 3 months. In parallel, monitoring of CMV infection in the lung was performed by examining BAL samples at routine visits for rejection surveillance (1, 3, 6 and 12 months after transplant) or whenever clinically indicated. CMV pneumonia was diagnosed on the basis of clinical signs and symptoms, imaging findings, presence of CMV DNA in the BAL or trans-bronchial biopsy specimen, and histological and immunohistochemistry examination of the biopsy specimen [25,30]. Based on a histological and immunohistochemical analysis of the biopsy specimens, lung infection was evaluated using the following classification scheme [25]: (i) absent, as assessed by immunohistochemistry; (ii) noncytopathic, with the presence of HCMV immediate-early antigens in the nuclei of the infected cells; (iii) cytopathic, in the presence of HCMV cytopathology (cytomegalic cells) involving endothelial cells, alveolar pneumocytes, ciliated cells or smooth muscle cells, but no lymphocyte infiltrates; (iv) pneumonia, with HCMV cytopathology and surrounding lymphocyte infiltrates.

In case of suspected gastrointestinal disease, endoscopic examination and virological, histological and immunohistochemical analysis of a biopsy specimen were performed.

CMV disease was classified as possible, probable or proven according to the “Disease Definitions Working Group of the Cytomegalovirus Drug Development Forum” [29].

### 2.3. Management and Treatment of CMV Infection or Disease

All the patients were managed with a pre-emptive approach, and no patient received antiviral prophylaxis for CMV infection. Pre-emptive antiviral therapy with GCV or VGCV for systemic CMV infection was initiated when the patient reached cutoffs previously established for antigenemia (for the 28 patients receiving antigenemia-guided pre-emptive therapy), that is, 100 pp65-positive/2 × 10^5^ leukocytes, and DNAemia (for all the other patients receiving DNAemia-guided pre-emptive therapy), that is, 300,000 DNA copies/mL blood [22]. The treatment was discontinued upon the first confirmed negative result. Lung CMV infection was treated when the viral load reached 100,000 DNA copies/mL of BAL [25]. CMV relapse, either systemic or pulmonary, was treated following the same criteria. Antiviral therapy consisted of i.v. ganciclovir (GCV) administration at the dosage of 5 mg/kg/bid or oral valganciclovir (VGCV) at the dosage of 900 mg/bid. CMV disease was treated with i.v. GCV at the dosage of 5 mg/kg/bid.

### 2.4. Immunosuppressive Management and Diagnosis of CLAD

Our chronic immune suppression protocol has undergone some changes over time. All patients transplanted between 2001 and 2007 were treated with a triple immunosuppressive regimen (cyclosporine, azathioprine and prednisone). Patients transplanted since January 2008 received a modified standard triple regimen (tacrolimus, mycophenolate mofetil and prednisone). In case of refractory acute rejection (AR), the patients were switched from cyclosporine to tacrolimus and from azathioprine to mycophenolate mofetil. In the presence of documented renal dysfunction, the patients were treated with low- dose tacrolimus plus everolimus [32]. 

All patients underwent surveillance and on-need bronchoscopies; the surveillance protocol has been reported elsewhere [33]. Biopsy-proven episodes of AR [34] were treated with steroid boluses and, in case of AR recurrence or persistence, with a standard anti-thymoglobulin course and a modulation of the immunosuppressive regimen. 

BOS diagnosis and severity grades were assessed according to published guidelines [35,36,37]. The CLAD subtype RAS was retrospectively re-classified for patients diagnosed before 2013, according to radiological (CT scan showing a pattern of persistent interstitial/upper lobe fibrosis) and functional criteria (persistent decline in forced expiratory volume in 1 s (FEV 1) of >20% compared to the best post Tx value and a decline in total lung capacity of >10% compared to baseline) [38,39]. In case of a BOS 0p or early RAS diagnosis, the patients were prescribed a 3-month course of chronic low-dose azithromycin. In case of a further decline consistent with a CLAD diagnosis, since 2003, the patients were referred to the Apheresis Unit for compassionate ECP treatment [40]. The severity of CLAD was classified according to the degree of functional impairment as described for BOS [35,36,37].

### 2.5. Statistical Analysis

Cumulative incidence curves were calculated by the Kaplan–Meyer method. To analyse the association between viral load and the development of CLAD, BOS or RAS, the patients were divided into two groups on the basis of peak viral load (patients with viral load above vs. patients with viral load below the median level), and the cumulative incidence curves were compared by the log-rank test. In addition, the patients were divided into three groups according to progressively higher peak viral load levels in the blood (<10^4^; 10^4^–10^5^; and >10^5^ copies/mL blood), and the cumulative incidence curves were compared by the log-rank test for trend. To analyse the association between CMV infection, acute rejection or CLAD with the factors sex, age, type of transplantation (single or double lung), induction treatment with anti-thymocyte globulin and use of mofetil mycophenolate in the immunosuppressive regimen, the χ^2^ test was used. The variables significantly associated with the incidence of CLAD in univariate analysis were subsequently analysed in a Cox regression multivariate model. Graphics and statistical analyses were performed with GraphPad Prism 6 (GraphPad Software, La Jolla, CA, USA) except for Cox regression, which was performed with NCSS 2004 (NCSS, Kaysville, Utah).

## 3. Results

### 3.1. Patients’ Characteristics

The characteristics of the patients analysed are shown in Table 1. Of the 129 patients analysed, 72 patients received a double-lung, 49 a single-lung, and 8 a heart–lung transplantation. The median age at transplantation was 54 (range 13–69) years (male/female ratio: 96/33); 12 patients were CMV-seronegative before transplantation (9 received the transplant from a CMV-seropositive donor), and 108 were CMV-seropositive, while the serostatus was unknown for 9 patients. Most patients received the standard triple immunosuppressive regimen with cyclosporin-A, azathioprine and steroid or tacrolimus, mycophenolate–mofetil and steroid. In addition, 29 patients received the induction therapy with anti-thymocyte globulin (ATG).

### 3.2. CMV Infection and Disease

CMV infection was detected in the blood of 113 (88%) patients after a median time of 27 (range: 4–293) days and in the BAL of 111 (86%) patients after a median time of 39 (range: 1–995) days, while in 8 (6%) patients CMV was never detected. Pre-emptive therapy was administered to 27 (21%) patients who reached the blood cutoff, to 28 patients who reached the BAL cutoff and to 16 (13%) patients who reached both the blood and the BAL cutoffs. Biopsy-proven CMV pneumonia was diagnosed in two patients, concomitantly with the detection of a CMV load above the cutoff level in the BAL (533,200 and 909,700 CMV DNA copies/mL). One patient showed signs of probable systemic syndrome and pneumonia (not biopsy-proven) in the presence of 684,200 CMV DNA copies/mL of blood and 1,489,200 CMV DNA copies/mL of BAL and simultaneous bacterial infection. In addition, three patients were treated for suspected retinitis (*n* = 1, ophthalmological diagnosis with no virological examination) and for possible gastrointestinal disease (*n* = 2) diagnosed by the presence of CMV DNA in biopsy specimens).

Overall, 52 (40%) patients did not develop a CMV infection or resolved it spontaneously (peak CMV DNA levels were 2930, (range: <50–228,400) copies/mL of blood, and 3250 (range: <50–86,850) copies/mL of BAL), whereas 71 (55%) patients received a median number of one (range: 1–3) course of pre-emptive GCV/VGCV for a median total duration of 28 (range: 9–191) days of treatment. Proven or probable CMV disease occurred in six (5%) patients and was healed after GCV treatment. The viral load levels in the blood and BAL of patients who resolved the infection spontaneously or who received the antiviral treatment are shown in Figure 1.

### 3.3. Acute Rejection, CMV Infection and CLAD

The cumulative incidence of CLAD was 38% at 5 years and 54% at 10 years. The cumulative incidence of BOS and RAS were 24% and 16% at 5 years, and 43% and 18% at 10 years, respectively (Figure 2).

Acute rejection occurred in 69 (53%) patients. As expected, CLAD occurred more frequently in patients who experienced acute rejection (Figure 3A), with a 10-year cumulative incidence of 63% vs. 26% in patients without acute rejection (*p* = 0.013).

The potential association between the cumulative incidence of CLAD and the level of CMV infection in the blood and BAL was analysed. The cumulative incidence of CLAD was significantly higher (*p* = 0.023) in patients with a high CMV load in the blood (i.e., above the median value) but was not significantly different in patients with a high or low CMV load in the BAL (Figure 3B,C).

In order to more deeply investigate the association between CLAD and CMV load, we divided the patients into three groups according to their CMV DNA peak level in the blood. The three CMV DNA intervals selected were the following: low (<10^4^ copies/mL), high (10^4^–10^5^ copies/mL) or very high (>10^5^ copies/mL) CMV load in the blood. CLAD occurred significantly different in the three groups (*p* < 0.001): the 10-year cumulative incidence of CLAD was 80% in patients with very high, 52% in patients with high and 35% in patients with low viral load (Figure 4A). Within the low viral load group (<10^4^ copies/mL), the 10-year cumulative incidence of CLAD was not different between 16 patients in whom CMV DNA was never detected (35%) and 32 patients with positive CMV DNA in the blood (38%).

When we analysed the occurrence of BOS and RAS separately (Figure 4B,C), we found that BOS was significantly more common (*p* = 0.024) in patients with a very high viral load (10-year cumulative incidence: 70%) compared to patients with a high or low viral load, in whom the cumulative incidence of BOS was similar (35% and 30%, respectively). On the contrary, RAS occurrence was more common (and at a similar rate) in patients with a very high and high viral load (10-year cumulative incidence: 33% and 23%, respectively) compared to patients with a low viral load (6%, *p* < 0.001).

We then analysed the potential association between CMV infection, acute rejection or CLAD with the following factors: sex, age (below or above the median value of 54 years), type of transplantation (single- or double-lung), induction treatment with anti-thymocyte globulin, and use of mofetil mycophenolate in the immunosuppressive regimen (Table 2). Single-lung transplantation was significantly associated with a higher CMV load and a higher incidence of CLAD, whereas the use of mycophenolate was associated with a lower incidence of both acute rejection and CLAD.

In order to evaluate the relative impact of the CMV load (divided into three intervals, as reported above), acute rejection, single-lung transplantation and the use of mycophenolate in the occurrence of CLAD, we analysed the above-mentioned factors in a Cox regression model. All the variables remained independently associated with the occurrence of CLAD (Table 3).

## 4. Discussion

The most frequent viral complication of LT recipients is CMV infection, which may affect patient’s outcome in the first months after transplantation. In the long-term period, chronic lung rejection (namely CLAD) is responsible for a reduced survival of LT recipients. In addition, CMV replication occurring in the early post-transplantation period has been suggested to be a potential trigger for the subsequent development of CLAD.

Two strategies are currently adopted to reduce the burden of CMV infection in patients receiving lung (as well as other organ) transplantation: pre-emptive therapy and universal prophylaxis. While both strategies should be equally effective in the prevention of CMV disease, universal prophylaxis could be more effective in preventing also the indirect effects of CMV infection, such as CLAD [22].

In this study, we analysed the impact of pre-emptive therapy in the prevention of CMV disease in 129 LT recipients receiving no anti-CMV prophylaxis. In addition, we explored the potential association between CMV infection and the occurrence of CLAD. In our cohort, CMV disease occurred in 5% of the patients (three cases of pneumonitis, two cases of gastrointestinal disease, one case of retinitis), whereas the risk for development of CLAD, which occurred in 38% of the patients at 5 years and 54% at 10 years, was associated with high levels of CMV load in the blood. On the other hand, the occurrence of CLAD was not associated with the levels of CMV in the lungs, determined by CMV DNA quantification in BAL samples.

Our retrospective analysis shows that pre-emptive therapy guided by the monitoring of CMV both in the blood and in the BAL is a highly effective strategy, able to prevent CMV disease in most patients. The occurrence of CMV disease and CMV pneumonia in our cohort (5% of patients) was lower than that reported in the literature for patients receiving 3–6 months of GCV/VGCV prophylaxis, in whom the frequency of CMV disease ranged from 13% to 57% [40,41,42,43,44]. Conversely, the frequency of CMV disease observed in our study was similar to that observed for patients receiving extended prophylaxis for 12 months or receiving prophylaxis for indefinite duration [41,42] and to that reported for patients receiving pre-emptive therapy post-prophylaxis [29,45]. In fact, in these groups of patients, CMV pneumonitis occurred in 1–12% cases. While breakthrough CMV disease in patients receiving prophylaxis is a rare event, a significant proportion of patients does develop late-onset CMV pneumonitis after the cessation of the prophylaxis, unless virological surveillance and pre-emptive therapy are implemented [29,45]. To reduce the risk for late-onset CMV disease after 3–6 months of prophylaxis, an extended (12 months) or indefinite course of prophylaxis has been proposed [41]. Concerns raised by the extended use of prophylaxis are relevant to the high costs of prolonged antiviral drug administration, to the bone marrow toxicity of GCV and VGCV and to the potential selection of antiviral resistance [46,47]. On the other hand, the logistical issues relevant to the strict virological surveillance necessary to guide pre-emptive therapy may overbalance the drawbacks of extended prophylaxis. Therefore, the prophylactic approach is preferred in many centres that cannot rely on a frequent virological monitoring of the transplanted patients. 

Besides its direct role as a major cause of pneumonitis or systemic syndromes, the possible role of CMV and CMV prevention in the development of CLAD has been debated for years [48].

In our population, the cumulative incidence of CLAD increased in patients with progressively higher peak levels of viral load in the blood. In line with our findings, a recent study conducted on lung transplant recipients receiving anti-CMV pre-emptive therapy analysed the number of CMV infection episodes instead of the peak viral load in the blood. The authors reported that patients with three or more episodes of CMV infection had a shorter CLAD-free survival than patients with no or 1–2 episodes of CMV infection [49]. On the other hand, in our population, the incidence of CLAD was not associated with the levels of CMV DNA in the BAL. 

Based on our results, it is not possible to define whether CMV infection has a causal role in triggering CLAD or if it is only a marker of the risk for the future development of CLAD, because it is not possible to define causal correlations with observational studies. Only a direct comparison in a randomized controlled trial of the two different management strategies (extended prophylaxis, which virtually prevents CMV replication, and pre-emptive therapy, which allows a certain degree of CMV replication), could verify whether CMV replication in the early post-transplantation period may be actually responsible for the subsequent development of CLAD.

Some studies observed an increased incidence of BOS in LT recipients developing pulmonary CMV infection or pneumonitis [43,44]. Conversely, other authors reported that treated CMV pneumonitis [50] and high levels of viral load in the BAL [45] were not associated with BOS development. The latter observations are in agreement with our findings. The lack of association between the levels of CMV in lungs and CLAD is against a direct role of CMV replication in CLAD.

Other arguments against the role of CMV as a trigger of CLAD arise from a large retrospective analysis [51]. This study did not find a lower incidence of BOS in seronegative patients with a seronegative donor (in whom CMV disease occurred only in 2% of the patients) compared to the other donor/recipient serostatus groups (in whom CMV disease occurred in 27–47% of the patients). Similarly, the patients receiving indefinite prophylaxis (therefore, with sustained suppression of CMV replication) had an incidence of BOS similar to that observed in patients who interrupted the prophylaxis [41], notwithstanding the great difference in CMV pneumonia observed in the two groups (2% vs. 57% of the cases, respectively). 

Finally, and most importantly, the incidence of CLAD observed in our cohort was not superior to that reported for patients in whom a sustained suppression of CMV replication was achieved by antiviral prophylaxis and the occurrence of CLAD approached 50% in 5 years [9], notwithstanding the fact that the pre-emptive approach allows a much higher degree of CMV replication before the initiation of the antiviral treatment than the prophylaxis.

We can speculate that CMV replication represents a prognostic parameter for the subsequent development of CLAD, rather than being a causal agent. The association between a high viral load in the blood and the occurrence of CLAD may be explained by the fact that a common trigger is at the basis of both CMV infection, which occurs early after transplantation, and CLAD, which occurs at a later time. In our study, both CMV infection and acute rejection were independently associated with CLAD. Another factor independently associated with CLAD was the transplantation of a single lung (which was also associated with a higher CMV load), while the use of mycophenolate was associated with a lower incidence of both acute rejection and CLAD. Animal and in vitro studies showed that an allo-reaction per se induces CMV reactivation and replication [52,53]. Therefore, a high level of CMV replication may be a consequence of a more intense allo-reaction, and the latter is actually responsible for both the high viral load reactivation and the subsequent development of CLAD. Thus, a high CMV load may be considered as an earlier prognostic indicator of the risk of CLAD.

In any case, were CMV a trigger of CLAD, a pre-emptive therapy cutoff of 10,000 CMV DNA copies/mL of blood would also be able to prevent the effect of CMV on CLAD in addition to CMV disease. In fact, in our cohort, patients with undetectable CMV DNA in the blood or CMV DNA copies ≤10.000/mL of blood had the same incidence of CLAD.

The major limitation of our study is the lack of a direct comparison with a control group receiving anti-CMV prophylaxis; therefore, we cannot conclude about the superiority or inferiority of the pre-emptive strategy adopted.

In conclusion, the pre-emptive approach is efficient in the prevention of CMV disease similarly to extended prophylaxis and does not induce a higher incidence of CLAD than prophylaxis. The decision to adopt pre-emptive therapy, extended prophylaxis or a hybrid strategy involving prophylaxis followed by pre-emptive therapy should depend upon a balance between the economic costs of prophylaxis and the logistic hurdles of pre-emptive therapy. In addition, patients developing a high CMV load in the blood should be considered at a higher risk for developing CLAD and should be addressed to appropriate prevention strategies.

## Figures and Tables

**Figure 1 microorganisms-10-02339-f001:**
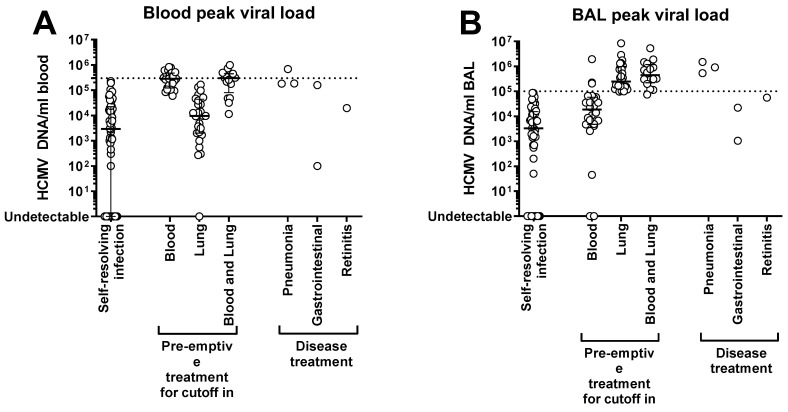
Peak levels of CMV DNA in (**A**) blood or (**B**) BAL of lung transplant recipients with self-resolving or no infection, pre-emptively treated for asymptomatic infection or treated for CMV disease. Each dot represents a patient. Median level and interquartile range are shown.

**Figure 2 microorganisms-10-02339-f002:**
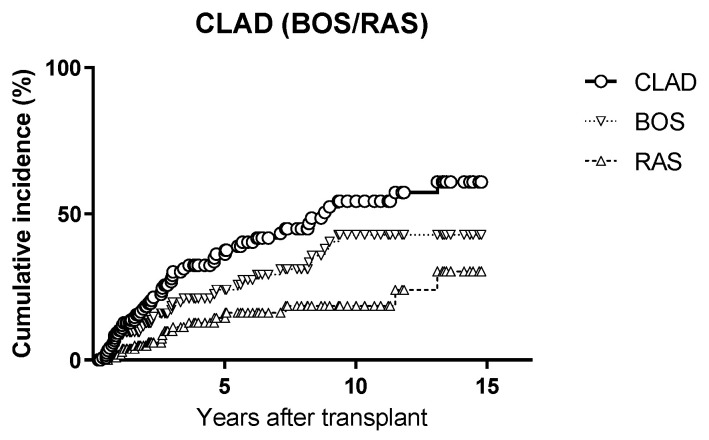
Cumulative incidence of CLAD, BOS and RAS.

**Figure 3 microorganisms-10-02339-f003:**
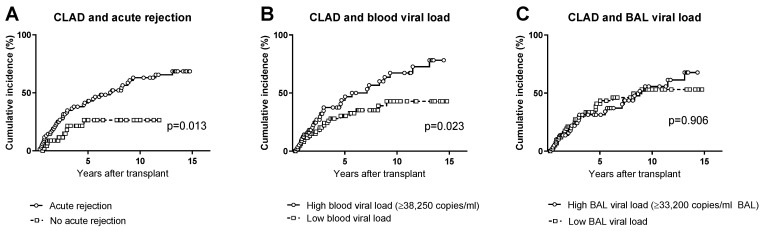
Cumulative incidence of CLAD in (**A**) patients who experienced acute rejection or no acute rejection; (**B**) patients who developed a CMV load in the blood above or below the median level; (**C**) patients who developed a CMV load in the BAL above or below the median level.

**Figure 4 microorganisms-10-02339-f004:**
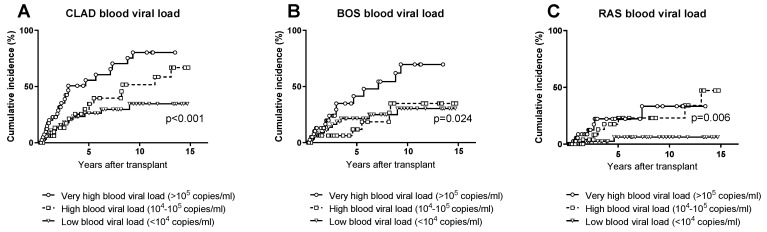
Cumulative incidence of (**A**) CLAD; (**B**) BOS; (**C**) RAS in patients who developed three progressively higher levels of CMV load in the blood.

**Table 1 microorganisms-10-02339-t001:** Characteristics of the 129 patients analysed.

Variable	Number (%)
Age, median (range)	54 (13–69)
Sex (M/F)	96/33 (74/26)
Type of transplantation	
Single-lung	49 (38)
Double-lung	72 (56)
Heart–lung	8 (6)
Donor (D)/Recipient (R) serostatus	
D positive/R positive	46 (36)
D negative/R positive	11 (8)
D unknown/R positive	51 (40)
D positive/R negative	9 (7)
D negative/R negative	3 (2)
D positive/R unknown	2 (2)
D unknown/R unknown	7 (5)
Use of anti-thymocyte globulin	29 (22)

**Table 2 microorganisms-10-02339-t002:** Factors associated with CMV load, acute rejection or chronic lung allograft dysfunction (CLAD).

	No (%) Patients with HCMV Load:						P (Chi^2^)	No. (%) Patients with Acute Rejection		P (Chi^2^)	10-Year Incidence of CLAD (%)	*p* (Log-Rank)
	Low		High		Very High							
Sex												
-M	36	(38)	26	(27)	34	(35)		54	(61)		29	
-F	9	(27)	11	(33)	13	(40)	0.555	15	(52)	0.360	71	0.083
Age												
-≤54	27	(40)	18	(27)	22	(33)		40	(64)		48	
->54	18	(29)	19	(31)	25	(40)	0.401	29	(54)	0.283	65	0.174
Type of transplant												
-Single-lung	11	(22)	14	(29)	24	(49)		22	(51)		73	
-Double-lung *	33	(42)	22	(28)	23	(30)	0.038	47	(64)	0.190	45	0.020
Induction with anti-thymocyte globulin												
-Yes	8	(28)	11	(38)	10	(34)		18	(72)		42	
-No	36	(38)	24	(25)	35	(37)	0.375	51	(55)	0.135	60	0.609
Use of mycophenolate **												
-Yes	30	(40)	18	(24)	27	(36)		30	(44)		48	
-No	15	(29)	18	(35)	19	(36)	0.316	37	(80)	<0.001	64	0.017

* Including heart–lung transplant recipients. ** Patients withdrawing from mycophenolate within the first 6 months were excluded from the analysis.

**Table 3 microorganisms-10-02339-t003:** Cox regression model to analyse the association of CMV load and acute rejection with CLAD.

Independent Variable	Risk Ratio (95% CI)	*p* Value
CMV load (low, high or very high)	1.7 (1.2–2.5)	0.007
Acute rejection	2.2 (1.0–4.7)	0.038
Single-lung transplantation	2.0 (1.1–3.6)	0.029
Use of mycophenolate	0.6 (0.3–1.0)	0.064

## Data Availability

The dataset used and analysed during the current study is available from the corresponding author on reasonable request.

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
