# Peer review of "Fifteen-Year Surveillance of LTR Receiving Pre-Emptive Therapy for CMV Infection: Prevention of CMV Disease and Incidence of CLAD"

_microorganisms, 2022, doi:10.3390/microorganisms10122339_

Round 1
Reviewer 1 Report
The manuscript by Piloni reported a study on 15-year surveillance of Lung transplantation recipients with/without pre-emptive treatment with ganciclovir/valganciclovir (GCV/VGCV) for CMV infection. Based on the data analysis, the authors try to determine if there is a relationship between CMV infection disease and the development of chronic lung allograft dysfunction (CLAD). The study itself and the presented data are interesting and important for Lung transplantation recipients. This reviewer has some concerns for authors to improve their manuscript before it can be accepted for possible publication.
1) First of all, the authors should pay more attention to the writing, organization, and logic of their paper. Careful proofreading and editing must be done before submission.
2) The authors should clarify the following issues.
a) relationship between Lung transplantation and CMV infection;
b) relationship between Lung transplantation and chronic lung allograft dysfunction;
c) relationship between CMV infection and chronic lung allograft dysfunction;
Based on the above results, the authors can then make conclusions. Otherwise, discussions may be invalid and confusing such as the style used in this manuscript.
3) Did the authors consider the PHM and FMH for all studied LTR? This should be carefully analyzed to exclude the original infection, such as CMV or other infections correlated with CLAD.
4) Rearrangement must be done for the discussion section.
Author Response
The manuscript by Piloni reported a study on 15-year surveillance of Lung transplantation recipients with/without pre-emptive treatment with ganciclovir/valganciclovir (GCV/VGCV) for CMV infection. Based on the data analysis, the authors try to determine if there is a relationship between CMV infection disease and the development of chronic lung allograft dysfunction (CLAD). The study itself and the presented data are interesting and important for Lung transplantation recipients. This reviewer has some concerns for authors to improve their manuscript before it can be accepted for possible publication.
1) First of all, the authors should pay more attention to the writing, organization, and logic of their paper. Careful proofreading and editing must be done before submission.
R: We amended the typos and revised the Discussion.
2) The authors should clarify the following issues.
- a) relationship between Lung transplantation and CMV infection;
- b) relationship between Lung transplantation and chronic lung allograft dysfunction;
- c) relationship between CMV infection and chronic lung allograft dysfunction;
Based on the above results, the authors can then make conclusions. Otherwise, discussions may be invalid and confusing such as the style used in this manuscript.
R: we thank the Reviewer for this observation and explained the relationship between lung transplantation, CMV infection and CLAD at the beginning of the Discussion
3) Did the authors consider the PHM and FMH for all studied LTR? This should be carefully analyzed to exclude the original infection, such as CMV or other infections correlated with CLAD.
R: The mentioned parameters are not considered for lung transplant recipients in our center. However, we analysed other factors potentially associated with CMV infection, acute rejection and CLAD (see new Table 2 and the modified Table 3 (previous Table 2).
4) Rearrangement must be done for the discussion section​
R: we revised the Discussion as mentioned above

Reviewer 2 Report
1) Mentioned, which software was employed to elaborate de graphics and the statistical analysis.
2) In table 1, indicate what dos ATG means.
3) For the Cox regression (table 2), the authors employed other variables to correct? or co-variables?
4) there was any clinical or demographic factor associated with acute rejection, CLAD, BOS and/or RAS?
Author Response
Reviewer 2
1) Mentioned, which software was employed to elaborate de graphics and the statistical analysis.
R: We reported the software employed for graphics and statistical analyses (GraphPad Prism 6 and NCSS 2004).
2) In table 1, indicate what dos ATG means.
R: we reported in extenso anti-thymocyte globuline instead of ATG
3) For the Cox regression (table 2), the authors employed other variables to correct? or co-variables?
R: In the modified Table 2 (now Table 3) we added type of transplantation (single or double lung) and use of mycophenolate in the Cox regression model
4) there was any clinical or demographic factor associated with acute rejection, CLAD, BOS and/or RAS?
R: In the new Table 2 we analysed clinical and demographic factors associated with CMV load, acute rejection and CLAD and added the significantly associated factors (type of transplantation and use of mycophenolate) in the Cox regression model (now Table 3).

Reviewer 3 Report
The nature of the study, which did not include a control group for preemptive therapy, limits the strength of the conclusions about the utility of this treatment strategy. However, within these limitations, the authors interpret their results in a reasonable manner and provide at least some modest new insights into the role of cytomegalovirus in the clinical course of lung transplant recipients.
In particular, the association of the two CLAD entities, BOS and RAS, with different levels of HCMV DNAemia is a relevant finding that raises the question of whether patients might benefit from a lower threshold for deciding on preemptive treatment.
To further improve their manuscript, the authors might consider looking for a common denominator for the level of HCMV DNAemia early after transplantation and the development of CLAD later in life. For example, the detailed treatment conditions mentioned in Section 2.4 could be examined for an association with the two outcomes.
Minor issues:
The authors should mention and discuss the findings in Bennett et al. Lung 2022 (Cytomegalovirus Infection Is Associated with Development of Chronic Lung Allograft Dysfunction)
line 38: typo ("heathy" should read "healthy")
line 81: typo ("costs. [29, 30]" should read "costs [29, 30].")
line 278: typo ("explred" should read "explored")
Author Response
Reviewer 3
The nature of the study, which did not include a control group for preemptive therapy, limits the strength of the conclusions about the utility of this treatment strategy. However, within these limitations, the authors interpret their results in a reasonable manner and provide at least some modest new insights into the role of cytomegalovirus in the clinical course of lung transplant recipients.
In particular, the association of the two CLAD entities, BOS and RAS, with different levels of HCMV DNAemia is a relevant finding that raises the question of whether patients might benefit from a lower threshold for deciding on preemptive treatment.
To further improve their manuscript, the authors might consider looking for a common denominator for the level of HCMV DNAemia early after transplantation and the development of CLAD later in life. For example, the detailed treatment conditions mentioned in Section 2.4 could be examined for an association with the two outcomes.
R: We thank the reviewer for the positive comments and analysed the potential association between sex, age (below or above the median value of 54 years), type of transplantation (single or double lung), induction treatment with anti-thymocyte globulin, use of mofetil mycophenolate in the immunosuppressive regimen, with CMV infection, acute rejection and CLAD (see new Table 2). In addition, we added in the Cox regression model the type of transplantation and use of mycophenolate among the independent variables (modified Table 3).
Minor issues:
The authors should mention and discuss the findings in Bennett et al. Lung 2022 (Cytomegalovirus Infection Is Associated with Development of Chronic Lung Allograft Dysfunction)
line 38: typo ("heathy" should read "healthy")
line 81: typo ("costs. [29, 30]" should read "costs [29, 30].")
line 278: typo ("explred" should read "explored")
R: We discussed the findings of the above mentioned article and corrected the typos.

Round 2
Reviewer 1 Report
This reviewer has no further comments on this revised manuscript.